# EpiBeds: Data informed modelling of the COVID-19 hospital burden in England

Christopher E. Overton[1,2,3,4☯*], Lorenzo Pellis[1,3,5☯], Helena B. Stage[1,6,7], Francesca Scarabel[1,3], Joshua Burton[8], Christophe Fraser[9,10,11], Ian Hall[1,2,3,5,12], Thomas A. House[1,2,3,5,8,13], Chris Jewell[14], Anel Nurtay[9], Filippo Pagani[1,15], Katrina A. Lythgoe[9,16]

1 Department of Mathematics, University of Manchester, Manchester United Kingdom, 2 Clinical Data Science Unit, Manchester University NHS Foundation Trust, Manchester, United Kingdom, 3 Joint UNIversities Pandemic and Epidemiological Research, https://maths.org/juniper/. Cambridge, United Kingdom, 4 Infectious Disease Modelling, All Hazards Intelligence, UK Health Security Agency, London, United Kingdom, 5 Alan Turing Institute, London, United Kingdom, 6 The Humboldt University of Berlin, Berlin, Germany, 7 The University of Potsdam, Potsdam, Germany, 8 Faculty of Biology Medicine and Health, Division of Informatics, Imaging and Data Sciences, University of Manchester, Manchester, United Kingdom, 9 Big Data Institute, Nuffield Department of Medicine, University of Oxford, Oxford, United Kingdom, 10 Wellcome Centre for Human Genetics, Nuffield Department of Medicine, NIHR Biomedical Research Centre, University of Oxford, Oxford, United Kingdom, 11 Wellcome Sanger Institute, Cambridge, United Kingdom, 12 Emergency Preparedness, Health Protection Division, UK Health Security Agency, London, United Kingdom, 13 IBM Research, Hartree Centre, Daresbury, United Kingdom, 14 CHICAS, Lancaster Medical School, Lancaster University, Lancaster, United Kingdom, 15 MRC Biostatistics Unit, University of Cambridge, Cambridge, United Kingdom, 16 Department of Biology, University of Oxford, Oxford, United Kingdom

☯ These authors contributed equally to this work.
* christopher.overton@manchester.ac.uk

**Data Availability Statement:** Code for simulating the EpiBeds model is available at: https://github. com/OvertonC/EpiBeds. The data used were provided through a data sharing agreement, and

## Abstract

The first year of the COVID-19 pandemic put considerable strain on healthcare systems worldwide. In order to predict the effect of the local epidemic on hospital capacity in England, we used a variety of data streams to inform the construction and parameterisation of a hospital progression model, EpiBeds, which was coupled to a model of the generalised epidemic. In this model, individuals progress through different pathways (e.g. may recover, die, or progress to intensive care and recover or die) and data from a partially complete patient-pathway line-list was used to provide initial estimates of the mean duration that individuals spend in the different hospital compartments. We then fitted EpiBeds using complete data on hospital occupancy and hospital deaths, enabling estimation of the proportion of individuals that follow the different clinical pathways, the reproduction number of the generalised epidemic, and to make short-term predictions of hospital bed demand. The construction of EpiBeds makes it straightforward to adapt to different patient pathways and settings beyond England. As part of the UK response to the pandemic, EpiBeds provided weekly forecasts to the NHS for hospital bed occupancy and admissions in England, Wales, Scotland, and Northern Ireland at national and regional scales.

unfortunately cannot be provided. Data similar to the SITREP and CPNS sources are available at, though with slightly different data definitions, are available at: https://coronavirus.data.gov.uk/details/download.

**Funding:** I.H. is supported by the National Institute for Health Research Health Protection Research Unit (NIHR HPRU) in Emergency Preparedness and Response. L.P., H.B.S. and C.E.O. are funded by the Wellcome Trust and the Royal Society (grant no. 202562/Z/16/Z). J.B. was supported by a Wellcome Trust Four-Year PhD Studentship in Basic Science (219992/Z/19/Z). C.J. is funded by the MRC (MR/V038613/1), EPSRC (EP/W011840/1, EP/R018561/1), and Wellcome (UNS73114). T.A.H. is supported by the Royal Society (grant no. INF\R2\180067). C.E.O., L.P., I.H., T.A.H. and F.S. are supported by the UKRI through the JUNIPER modelling consortium [grant number MR/V038613/1]. K.A.L. is funded by the Wellcome Trust and The Royal Society (107652/Z/15/Z) and the Li Ka Shing Foundation. I.H. is supported by the National Institute for Health Research Policy Research Programme in Operational Research (OPERA). The funders had no role in study design, data collection and analysis, decision to publish, or preparation of the manuscript.

**Competing interests:** The authors have declared that no competing interests exist.

## Author summary

COVID-19, the disease caused by SARS-CoV-2, leads to a high proportion of cases requiring admission to hospital. Coupled with the high burden of infections worldwide, this put substantial pressure on healthcare systems. To enable public health systems to cope with the high levels of demand, forecasting models are vital. These models enable public health managers to plan their workloads accordingly. Here, we developed EpiBeds, which combines an epidemic model with a model for patient flow through hospitals. By fitting this model to data from England, EpiBeds has been used to provide short-term forecasts of hospital admissions and bed demand weekly throughout the COVID-19 pandemic. In this paper, we describe the motivation behind the structure of EpiBeds, how the model is fitted to data, and report the estimates of the key parameters throughout the pandemic. We then evaluate the performance of EpiBeds by comparing generated forecasts to future data points, finding good agreement between the forecasts and data.

## 1. Introduction

An important component of the UK response to the COVID-19 pandemic was the short-term prediction of hospital and critical care bed use for planning purposes. As part of this response, we developed EpiBeds, a minimally complex compartmental model tailored to data available on hospital flow and the natural history of disease progression that was available at the time. We fitted EpiBeds to four data streams: daily hospital admissions, daily hospital prevalence, daily intensive care unit (ICU) prevalence, and daily deaths in hospital, enabling us to make short-term projections of hospital and ICU bed demand, and to estimate the basic reproduction number, $R$. These predictions were used to support the resource management of the National Health Service of England, nationally and separately for each English region, and the other Devolved Administrations in the UK.

Forecasting models for hospital occupancy typically assume that individuals in certain bed types have the same waiting time distribution in that bed type regardless of outcome [1,2]. However, analysis of hospital line-list data showed that outcome was a major determinant of lengths of stay along the hospital pathway [3], and therefore in EpiBeds we defined hospital compartments not only by the current status of the patient (e.g. in critical care), but also their outcome (e.g. will recover). Defining multiple compartments was necessary since compartmental models typically require all individuals within a single compartment to have the same waiting time distribution. By doing so, we were able to maximise the information in the available data whilst minimising the complexity of the model. We reduced the number of unknown parameters using high-resolution individual-level data for a subset of hospitalised patients in England, to estimate the length of stay in each hospital compartment (conditional on the progression to each possible following stage) of the EpiBeds model.

Since hospitalisation data reflect background incidence, in addition to generating forecasts, EpiBeds enabled us to approximate the transmission rates in the background epidemic, and hence to provide real-time estimates of the instantaneous growth rate and effective reproduction number, published weekly by the UK Government. When policy was known to have changed recently or to be about to change, often multiple scenarios were submitted in addition to the projections (which assumed no change in transmission from the day of the projection), with a range of fixed values for the reproduction number from the date of the policy change.

Here we describe the motivation behind the structure of EpiBeds, including the structure of the model and the baseline parameter estimates. We then describe the model fitting procedure,

outlining how the background epidemic is captured and how the model is adapted to capture changes in patient dynamics. We then illustrate the performance of the model over the first and second waves, and report posterior estimates of the key epidemiological parameters. We end with an evaluation of model performance across the first and second waves of the pandemic. The relative simplicity of EpiBeds makes it more transparent than more complex models [4–6] about:blank, and unlike other models enables us to estimate the probability of moving along different hospital pathways. The simplicity enables issues in the model fitting to be easily identified and corrected, highlighting when relationships between the underlying data streams change or the model assumptions are violated. Additionally, with the particularly sparse data at the start of the pandemic, the minimally complex design ensured minimal assumptions were required when fitting the model. The flexibility of its construction and parameterisation also means it can easily be adapted to provide accurate short-term forecasts for different countries and healthcare systems, and potentially other pathogens, with the model structure tailored to the observed data.

## 2. Results

### 2.1. Estimates of hospital length of stay distributions

To inform the EpiBeds model structure, we first analysed the detailed COVID-19 Hospitalisation in England Surveillance System (CHESS) and Severe Acute Respiratory Infection (SARI) datasets (see Section SM.1.1 in S1 Supplementary Material) to identify the most relevant hospital pathways and to estimate the distributions of the time individuals spent along each step of these hospital pathways. We classed patients using five states: Hospitalised (not been to ICU), in Critical care (ICU), Monitored (discharged from ICU but still in hospital), Recovered, and Deceased. After hospital admission, patients are either discharged, admitted to ICU, or die (without entering ICU), and from ICU individuals may go on to be discharged from ICU (but remain in hospital in the monitored state) or die. We then estimated the distributions of the time individuals take for each transition (hereafter referred to as "length of stay" or "delay distribution", with the former preferred for in-hospital events and the latter preferred for out-of-hospital events), in particular: hospital admission to ICU admission, ICU admission to ICU discharge, ICU admission to death, ICU discharge to hospital discharge, hospital admission to death and hospital admission to hospital discharge. For hospital admission to death and hospital admission to discharge, we only considered patients who are not admitted to ICU, to prevent overlap with the ICU-related pathways.

Our aim was to produce a set of ordinary differential equations (ODEs) that best describe hospital progression. We therefore assumed length of stay distributions were gamma distributed, so that they could be approximated by Erlang distributions (see Section SM.1.2 in S1 Supplementary Material). Since treatment policies and practices, and patient demographics, are likely to have changed over time, we estimated the waiting time distributions separately for the first (1st March 2020 to 15th September 2020) and second (1st August 2020 to 31st December 2020) waves in the UK (Table 1), with monthly cumulative estimates given in Table D in S1 Supplementary Material. Our estimates are consistent with previous results for length-of-stay distributions (Fig A in S1 Supplementary Material), particularly findings for the UK, Europe and Japan ([7–16]). Note that the first and second wave periods have some overlap, as some historic data was needed to fit the second wave.

Comparing the first wave to the second wave, we observe substantial changes in the lengths of stay on ICU. The length of stay from entering ICU to dying slightly increased between the two waves, whilst the length of stay from entering ICU to leaving ICU decreased by a factor of two. Similarly, the length of stay from leaving ICU to discharge decreased by a factor of two.

**Table 1. Gamma distributed length of stay for different events in hospital, estimated using the CHESS/SARI data.** Brackets indicate 95% confidence intervals (generated through parametric bootstrapping).

| Length of stay | Wave[1] | Mean | Standard deviation | N |
|---|---|---|---|---|
| Hosp to ICU | Wave 1 | 2.79 (2.71, 2.87) | 3.30 (3.20, 3.41) | 6254 |
| | Wave 2 | 2.70 (2.61, 2.79) | 2.96 (2.83, 3.07) | 3830 |
| ICU to death | Wave 1 | 11.84 (11.43, 12.25) | 9.74 (9.35, 10.20) | 2268 |
| | Wave 2 | 15.33 (14.50, 16.08) | 12.38 (11.52, 13.18) | 837 |
| ICU to monitoring | Wave 1 | 15.93 (15.39, 16.52) | 16.97 (16.30, 17.64) | 3642 |
| | Wave 2 | 8.57 (8.18, 8.98) | 7.51 (7.05, 7.96) | 1348 |
| Monitoring to recovery | Wave 1 | 11.85 (11.39, 12.29) | 11.93 (11.37, 12.44) | 2602 |
| | Wave 2 | 6.45 (6.04, 6.90) | 6.58 (6.09, 7.09) | 945 |
| Hosp to recovery (no ICU) | Wave 1 | 9.37 (9.14, 9.60) | 9.68 (9.41, 9.96) | 6312 |
| | Wave 2 | 10.02 (9.66, 10.42) | 9.89 (9.43, 10.35) | 2462 |
| Hosp to death (no ICU) | Wave 1 | 8.93 (8.58, 9.27) | 7.81 (7.44, 8.16) | 2144 |
| | Wave 2 | 12.16 (11.43, 12.92) | 10.40 (9.59, 11.23) | 674 |

[1]Wave 1 includes dates 1st March 2020 to 15th September 2020 and wave 2 dates from 1st August 2020 to 31st December.

There are various potential drivers for this. First, treatment changes could have reduced the length of time patients require critical care treatment, and prolonged the time until death. Second, younger patients, who were more common in the second wave, take less time to recover and longer to die. The lengths of stay without ICU does not show the same drop in the time to recovery as seen in ICU, but has a similar increase in the time to death, possibly because of improved quality of treatment.

## 2.2. Construction of a compartmental model informed by hospital flow data

Informed by the estimated length of stay distributions (Table 1), we constructed a compartmental model describing the progression of individuals through the hospital pathways (Fig 1). To account for considerable differences in the duration of different hospital transitions even from the same state, we divided individuals into compartments both in terms of their current status (e.g., Hospitalised or Critical care) and in terms of their future outcome (e.g., will recover, will die). This approach requires more parameters than the more common approach based on competing hazards but is more flexible (resulting in more general phase-type sojourn times in each state) and can be directly parameterised with the available data. Since the mean and standard deviation of the estimated lengths of stay in each compartment are similar, this indicates the gamma distributions are approximately exponential (shape parameter 1). Therefore, flows between hospital compartments are suitably described by constant transition rates (equal to the inverse of the mean of the exponentially distributed sojourn time in the compartment, see Section 4.1 [17]). The resultant hospital flow is shown by the red and orange compartments in Fig 1.

Since the infectious burden in the population determines the rate at which cases will be admitted to hospital, we also used compartments to describe the process of infection in the general population, based on an SEIR (Susceptible Exposed Infectious Recovered) model structure. Symptomatic individuals therefore go through three states of infection, Exposed (but not yet infectious), Infectious (but not yet symptomatic), and Late infection (infectious and symptomatic), with a proportion of symptomatic individuals requiring hospitalisation ($L_H$) and the other proportion recovering naturally ($L_R$). The latter distinction is motivated by

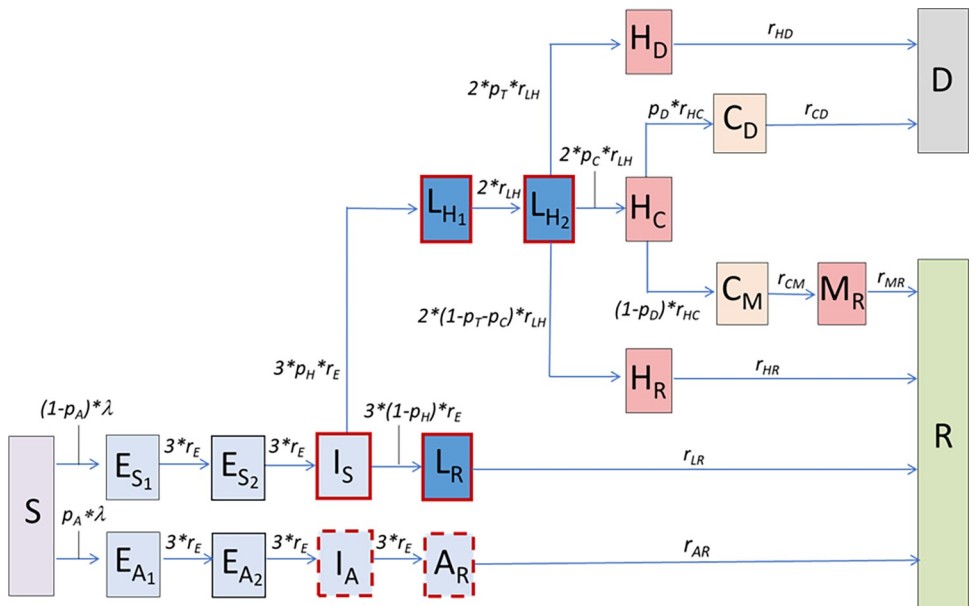

**Fig 1. Schematic representation of the EpiBeds model.** The construction of the compartmental model was informed by available data. EpiBeds is implemented as a set of ordinary differential equations (ODEs), with one state variable per compartment representing the absolute number of individuals in it. Arrows describe flow between compartments, which occurs at constant rate. Blue compartments indicate infected individuals who are not hospitalised, with a dark and light blue distinction, respectively, for individuals with and without symptoms, while red compartments indicate hospitalised individuals and orange compartments individuals in critical care. The compartments with a red border contain infectious individuals, with a dashed border denoting an infectivity reduced to 25% of that of the other infectious compartments; once hospitalised, it is assumed individuals no longer contribute to the community epidemic. For states in which the waiting times are not exponentially distributed (e.g. Exposed) we use multiple identical compartments enabling us to approximate gamma-distributed waiting times by using Erlang distributions. All variables, rates, and probabilities are described in Tables 2 and 3. The force of infection λ depends on the numbers in the infectious compartments (Section 4.1).

the fact that the processes of biological recovery and hospital seeking behaviour are conceptually different, hence involving different progression rates: for an infected individual, the time to recovery reflects the natural course of a non-severe infection, while the time to hospital admission is driven by hospital seeking behaviour, current policy, and health-care logistic availability. A proportion of individuals are assumed to remain asymptomatic throughout infection; these individuals follow an infection pathway that is distinct from, but mimics, that of symptomatic individuals.

The structure for the generalised epidemic was constructed to reflect delay distributions from the literature, using constant rates to represent exponentially distributed sojourn times, and sequences of compartments to represent gamma-distributed sojourn times (more details in Section 4.1). This is known as "linear chain trickery" and is a way of representing gamma-distributed sojourn times by using the Erlang distribution. Hence, to describe a gamma-distributed incubation period (i.e., the time from infection to symptom onset) with mean 4.85 days and shape parameter 3 [18], we used three subsequent compartments ($E_1$,$E_2$, I) with identical constant rates between them, with mean sojourn time 1.6 days in each compartment [17]. This assumes pre-symptomatic transmission of 1.6 days, which is roughly consistent with literature estimates that show most pre-symptomatic transmission occurs in the two days prior to symptom onset [19]. The delay between symptom onset and hospitalisation is gamma distributed with shape parameter approximately equal to two [18], and we therefore used two compartments for late-infection symptomatic individuals who will be hospitalised ($L_H$). For cases

that recover without hospitalisation, in the absence of better data on the duration of infectivity since symptom onset, we made the parsimonious choice of a single late infection compartment with an exponentially distributed length of stay with mean 3.5 days, such that the overall period during which an individual is actively infectious (I plus the L compartments) is consistent with the 5-day mean generation time estimated in [20]. The resultant compartmental model is illustrated in Fig 1, with the state variables and parameters described in Tables 2 and 3. The equations are reported in Section 4.1.

We assumed only non-hospitalised infectious individuals contribute to new infections, with asymptomatic individuals less infectious than individuals who are pre-symptomatic or symptomatic. Due to behavioural changes, changes in test specificity, and the possibility that asymptomatic cases may correspond to individuals who simply have a long incubation period, identification of the relative infectivity of an asymptomatic case is challenging. We assume relative infectivity of 25%, based on [30,31]. We assume that asymptomatic cases make up 55% of infections, which we determined by adjusting age specific estimates of the asymptomatic rate to the age distribution in England [24]. Although infections from hospitalised patients could have an effect on the overall epidemic, most notably with health care workers as transmission links, detailed genetic data are required to characterise this process [32]. We also assume that nosocomial cases do not substantially alter hospital flow, i.e., upon testing positive nosocomial patients follow similar pathways to community-acquired cases. In the hospital admissions data, we either count patients from admission (if they were tested in the community) or from the date of their first positive swab result (if they were tested in hospital). This second cohort will include all nosocomial cases, who we treat as being admitted from the community.

## 2.3. Model fitting

**2.3.1. Procedure.** We fitted EpiBeds to English data (SITREP—NHS situation report and CPNS—COVID-19 Patient Notification System) using a Bayesian MCMC approach (Section 4.2). When fitting to data, we considered waves one and two independently in order to capture temporal changes in the hospital dynamics. Since there were substantial parameter changes between the first and second wave, when fitting the second wave we used admissions for the whole time-series combined with beds, ICU, and deaths data only from 1st August 2020 onwards. This enabled the probabilities to be fitted to the second wave independently of the first wave, while still accounting for the depletion of susceptibles throughout the first wave and reasonable initial conditions for all variables at the start of the second wave.

To reduce the number of free parameters, we used the average waiting times in each hospital compartment for each wave estimated from the CHESS/SARI data (Table 1), and previously

**Table 2. State variables for the compartmental model.**

| State variable | Description |
|---|---|
| S | Susceptible |
| $E_A$, $E_S$ | Exposed–will stay asymptomatic, become symptomatic |
| $I_A$, $I_S$ | Infectious–will stay asymptomatic, become symptomatic |
| $A_R$ | Asymptomatic–will recover |
| $L_R$, $L_H$ | Late infection (symptomatic)–will recover, be hospitalised |
| $H_R$, $H_C$, $H_D$ | Hospitalised–will recover, enter critical care, die without entering critical care |
| $C_M$, $C_D$ | Critical care–will be monitored before recovery, die |
| $M_R$ | Monitored–will recover |
| R | Recovered |
| D | Deceased |

**Table 3. Parameter variables and values for the compartmental model.**

| Parameter variable | Description | Fixed parameter or prior distribution | Literature range | References |
|---|---|---|---|---|
| $r_E$ | Rate of transition through early stage infectious classes ($E_S$, $I_S$, $E_A$, $I_A$) | 1/4.85 | 1/4.85 | [18] |
| $r_{AR}$ | Rate of transition from late stage asymptomatic ($A_R$) to recovered (R) | 1/3.5 | (see text) | [20] |
| $r_{LR}$ | Rate of transition from late stage symptomatic ($L_R$) to recovered (R) | 1/3.5 | (see text) | [20] |
| $r_{LH}$ | Rate of transition from late stage severely symptomatic ($L_H$) | 1/5.2 | 1/5.2 | [18] |
| $r_{HR}$ | Rate of transition from hospital admission ($H_R$) to recovered (R), without ICU | 1/9.37 –Wave 1<br>1/10.02 –Wave 2 | 1/6.1 | [21], Table D in S1 Supplementary Material |
| $r_{HC}$ | Rate of transition from hospital admission ($H_C$) to ICU ($C_M$, $C_D$) | 1/2.79 –Wave 1<br>1/2.70 –Wave 2 | 1/4 to 1/1.5 | [7,8,21,22], Table D in S1 Supplementary Material |
| $r_{HD}$ | Rate of transition from hospital admissions ($H_D$) to death (D), without ICU | 1/8.93 –Wave 1<br>1/12.16 –Wave 2 | 1/9.8 to 1/7.5 | [23], Table D S1 in Supplementary Material |
| $r_{CM}$ | Rate of transition from critical care admission ($C_M$) to step down ($M_R$) | 1/15.93 –Wave 1<br>1/8.57 –Wave 2 | 1/16.8 to 1/12 | [11,21], Table D in S1 Supplementary Material |
| $r_{CD}$ | Rate of transition from critical care admission ($C_D$) to death (D) | 1/11.84 –Wave 1<br>1/15.33 –Wave 2 | 1/17 to 1/7 | [7,11,13], Table D in S1 Supplementary Material |
| $r_{MR}$ | Rate of transition from step down ($M_R$) to discharge (R) | 1/11.85 –Wave 1<br>1/6.45 –Wave 2 | 1/7 | [11], Table D S1 in Supplementary Material |
| $p_A$ | Proportion of infected individuals that will be asymptomatic | 0.55 | 0.179 to 0.972 | [24–28] |
| $p_H$ | Proportion of symptomatic individuals that will be hospitalised | 0.08 | 0.036 to 0.155 | [21,22,29] |
| $p_C$ | Proportion of hospitalised individuals that will enter critical care | Uninformative prior | 0.091 to 0.485 | [8,14,21,22,29] |
| $p_T$ | Proportion of hospitalised individuals that will die without entering critical care | Uninformative prior | 0.316 | [8] |
| $p_D$ | Proportion of individuals in critical care that will die | [1]Wave 1: 0.357 (0.319–0.384)<br>Wave 2: 0.287 (0.265–0.321) | 0.4 to 0.453 | [8,11] |

[1]We used a strongly informative Normal prior distribution obtained from SARI data, with mean and 95% CI shown.

published estimates for disease parameters (Table 3), as fixed model parameters. For the remaining parameters (Table 3) we used uninformative priors with the exception of the probabilities of death if in ICU ($p_D$). This is because the data on deaths and recoveries do not distinguish whether individuals have transitioned to ICU or not, and hence are both affected simultaneously by a combination of $p_C$ and $p_D$ (through ICU) and $p_T$ (without passing through ICU) thus making these three parameters only weakly identifiable (at best) if at least one of them is not constrained separately. For $p_D$ we used a strongly informative Normal prior distribution with a mean and 95% CI estimated from CHESS/SARI data for wave one at 35.7% (31.9%, 38.4%) and for wave 2 at 28.7% (26.5%, 32.1%). Obtaining similar priors for $p_T$ and $p_C$ (probability of entering ICU if hospitalised) was not possible due to insufficient and geographically uneven coverage in the data, causing problems in both power and representativeness.

The background epidemic is driven by a transmission rate, that represents the total infectious pressure exerted by a symptomatic infectious individual. This parameter collates contact behaviour, transmission probability of contacts and strength of contacts into a single parameter. On an individual level, this does not provide accurate information about the transmission dynamics, but on a population level aggregating all of these into a single parameter is a simple way to represent the average transmission dynamics in the population.

To model the background epidemic, we need to estimate the value of this transmission parameter. We cannot assume this transmission rate is constant, because there are large changes in this parameter across the pandemic, for example due to behavioural changes, implementation of control policies, and circulation of different variants. However, we do not want to add too many different values, as this risks overfitting noise in the data rather than genuine changes in transmission. To capture these large changes, we assumed the transmission rate was piecewise constant with pre-selected change points that generally correspond to large policy changes:

- 13th March 2020 (visible change in hospitalisation trend, possibly due to media-driven behavioural changes or inaccuracies in recording early hospitalisation data),

- 24th March 2020 (beginning of a UK-wide lockdown),

- 11th April 2020 (visible change in trend towards the end of lockdown),

- 15th August 2020 (visible rise in hospital admissions),

- 6th September 2020 (visible change in trend),

- 14th October 2020 (Merseyside first area in England to enter "tier 3" restrictions),

- 5th November 2020 (England-wide second lockdown),

- 18th November 2020 (indicated by an increase in infections due to the rise of the B.1.1.7 variant–now called Alpha–in England and potentially increasing social interactions, this also encompasses any transmission changes after lifting the second lockdown on 2 December 2020).

In addition, we included change points three weeks before the final data point, unless a major intervention was already present within the last three weeks. This translates in additional transmission rate changes on:

- 25th August 2020, when producing the fit to the entire first wave (Fig 2),

- 10th December 2020, when producing the fit to the entire second wave (Fig 3).

We refer to the periods during which transmission rates are assumed to be constant as constant-transmission intervals. Although further changes in transmission rates could have been added, this risked overfitting to noise in the data rather than genuine transmission trends. For a full description see Supplementary Methods in S1 Supplementary Material.

**2.3.2. EpiBeds captures the dynamics of the first and second waves in England.**   EpiBeds performed well in capturing the dynamics of both the first and second waves (Figs 2 and 3). For the first wave, the model fits admissions and hospital beds particularly well (low overdispersion of data around the average model prediction), whereas ICU occupancy and deaths required high overdispersion to capture the data. This is driven by multiple factors including: data quality issues between data streams at the start of the first wave; a large shift in the age distribution of admissions from frailer older people in the spring to younger people with low mortality risk in the summer; and changes in treatment which likely altered outcome probabilities.

For the second wave (Fig 3), there is better agreement among the data streams, due to more consistent reporting of data by the hospital trusts, less demographic shift in hospital admissions, and less dramatic changes in treatments, compared to the first wave. Although EpiBeds links all four data streams well during this period, there was a sharper increase in ICU admission during September 2020 than the model captured. During this period, admissions were

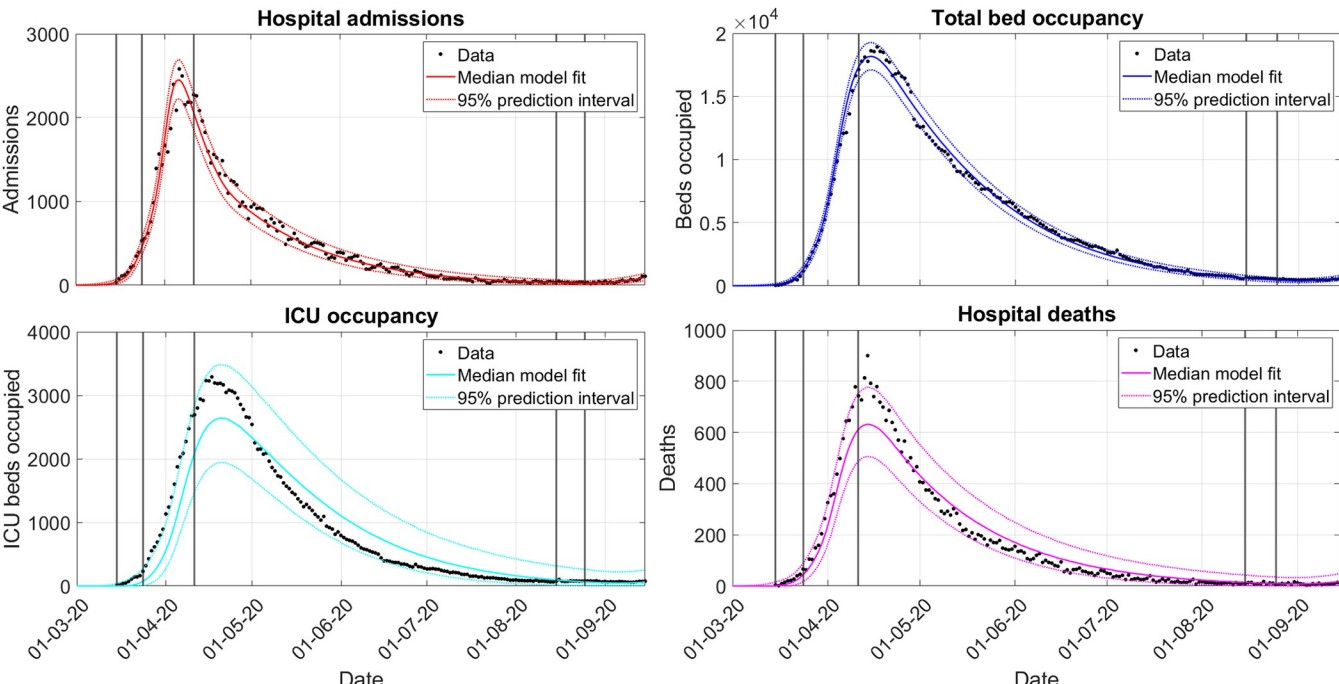

**Fig 2.** First-wave model fit to admissions (red), beds (blue), ICU beds (cyan) and deaths (magenta). Vertical lines indicate when transmission rate changes are added to EpiBeds (see Section SM.1.3.3 in S1 Supplementary Material). Note that there is a delay between transmission changing in the community and its effect being observed in the hospital data, so visual inflections in the model trend occur after the transmission change point. The 90% prediction interval was calculated by generating, for each parameter posterior sample, a new potential realisation of the data and then taking the 5 and 95 quantiles of the set of realised data at each time point. All dates are given as day/month/year.

concentrated in the relatively young, with severely ill younger patients more likely to visit ICU rather than be treated on the ward compared to older patients, since younger patients have more favourable ICU outcomes. As the epidemic spread through the community, the age distribution became relatively stable, corresponding to a slowdown in the ICU admission rate from October. Due to data quality issues in the early admissions data, we changed the data definitions used between the first and second waves slightly (see Section SM.1.1 in S1 Supplementary Material), resulting in higher admissions in the data used when fitting the second wave. Since for the second wave we only fitted the other three streams from 1st August 2020 onwards, these data quality issues no longer affect the performance of EpiBeds when linking the four data streams.

**2.3.3. The probabilities of dying, with and without ICU, declined significantly between waves.** Through the model fitting we obtained posterior estimates for the free parameters (see Table SM.1.2 in S1 Supplementary Material for the list of parameters—estimates are only reported for those with epidemiological significance, posterior distributions for all parameters can be found on Github [33]), including the outcome probabilities $p_D$, $p_T$, and $p_C$ (Table 4). These outcome probabilities were assumed to be constant throughout each wave and are presented only at the end of wave one (15th September 2020) and wave two (31st December 2020), to highlight the difference between waves (Table 4). Since we used strongly informative priors for $p_D$, the posterior estimates of $p_D$ generated through MCMC remained close to the prior, though we did observe a significant reduction between waves one and two (from 34% to 30%). The estimated probability of being admitted to ICU ($p_C$) remained relatively constant throughout 2020 at ~13%, in line with previous estimates [3,19,21]. In contrast, the probability of dying without entering ICU ($p_T$) dropped by more than 25% between the two waves, from

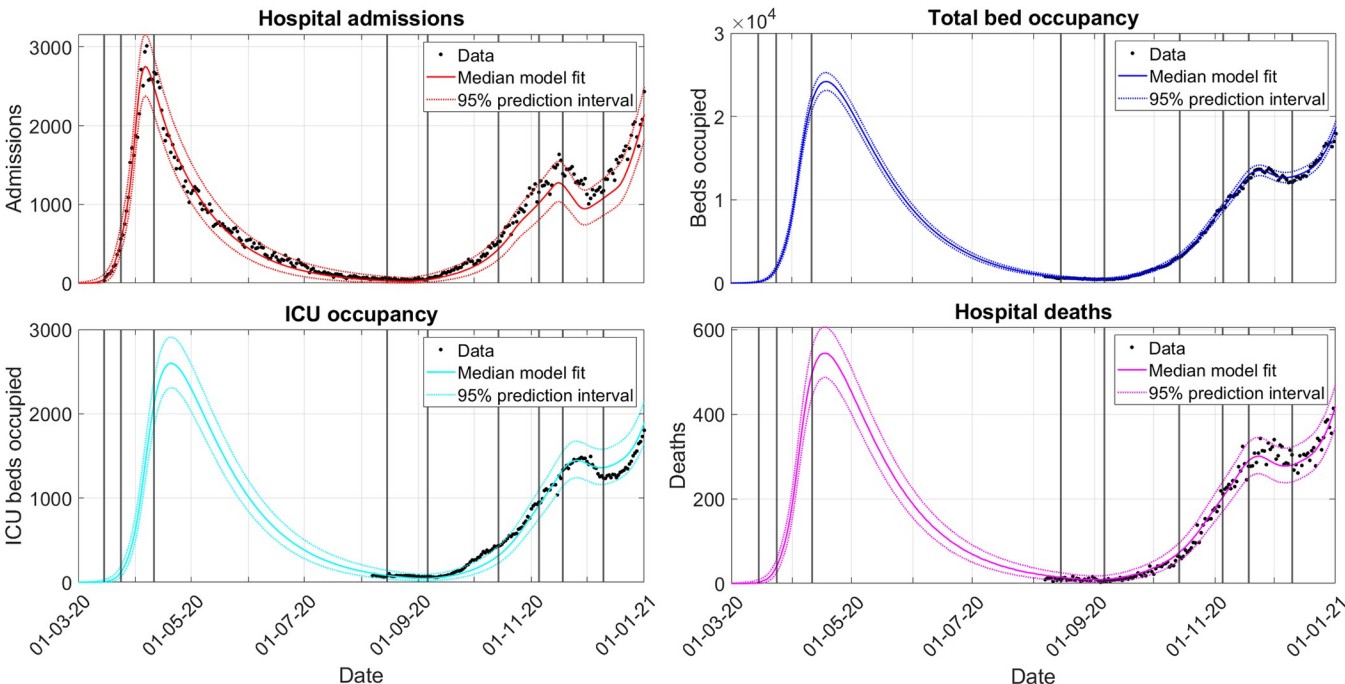

**Fig 3.** Second-wave model fit to admissions (red), beds (blue), ICU beds (cyan) and deaths (magenta). Admissions data were fitted starting from 1st March 2020, while the other three data streams were fitted starting from 1st August 2020. Vertical lines indicate when transmission rate changes are added to EpiBeds (see Section SM.1.3.3 in S1 Supplementary Material). Note that there is a delay between transmission changing in the community and its effect being observed in the hospital data, so visual inflections in the model trend occur after the transmission change point. The 90% prediction interval was calculated by generating, for each parameter posterior sample, a new potential realisation of the data and then taking the 5 and 95 quantiles of the set of realised data at each time point. All dates are given as day/month/year.

32% to 23%. These posterior estimates are consistent with the range of estimates from the literature (Fig C in S1 Supplementary Material).

In line with other published estimates [3,22,29], we estimated 13% of COVID-19 patients were admitted to ICU, during both the first and second waves in England. The proportion of patients surviving on ICU improved over time, with 34% mortality during the first wave and 30% during the second wave. An even stronger reduction in mortality occurred outside the ICU, with 32% of admissions dying without ICU during the first wave and 23% during the second wave. This reflects the change in the age distribution of cases and potential improvements in treatments. Given only 13% of admitted patients went to ICU, the vast majority of deaths

**Table 4. Posterior estimates for hospital pathway proportions during 2020.** Note that these periods overlap because some historic data were needed to fit the second wave.

| Parameter | Before 15th September | After 1st August |
|---|---|---|
| Proportion of hospitalised individuals that will enter critical care, $p_C$ | 0.125 (0.119, 0.130)[1] | 0.127 (0.123, 0.129) |
| Proportion of hospitalised individuals that will die without entering critical care, $p_T$ | 0.317 (0.305, 0.329) | 0.234 (0.229, 0.240) |
| Proportion of individuals in critical care that will die, $p_D$ | 0.344 (0.318, 0.372) | 0.296 (0.270, 0.321) |

[1]Parentheses indicate 90% credible intervals

occurred outside of ICU (about 89% and 84% of deaths during the two waves). In most cases these were frail individuals for whom ICU was unsuitable.

**2.3.4. Reproduction numbers fluctuated considerably during 2020.** Using the transmission rates determined from EpiBeds, we estimated two types of reproduction numbers: the control reproduction number $R_c(t)$ and the effective reproduction number $R_e(t)$ [6]. The control reproduction number $R_c(t)$ is the average number of new infections generated by an average infection started at time $t$, in the absence of population immunity, assuming the transmission rate does not change (e.g. due to policy changes affecting physical distancing) from its value at time $t$. The basic reproduction number $R_0$ is then given by $R_c(t)$ before the first intervention reduces transmission by limiting the "natural" (i.e. pre-pandemic) population contact patterns. The effective reproduction number, $R_e(t)$ (also denoted $R_t$), describes the average number of new infections generated by an average infection started at time $t$, taking into account population immunity. This can be obtained by multiplying $R_c(t)$ by the susceptible fraction of the population at time $t$.

We calculated $R_c(t)$ and $R_e(t)$ (Section 4.1) for each constant-transmission interval, using estimates of the transmission rate obtained when fitting only to data obtained during the first wave, or data from both waves (Table 5). The longer the interval during which the transmission rate is assumed to be constant, the smaller the uncertainty. Moreover, the estimates of $R_e(t)$ that are obtained when only fitting the first wave are constrained by all four data streams, whilst the first wave $R_e(t)$ estimates obtained when fitting to the second wave are only constrained by the hospital admissions, resulting in the slightly different estimates.

Although $R_c(t)$ is proportional to the transmission rate and hence is constant throughout each constant-transmission interval, as the proportion of susceptibles changes continuously over time, so does $R_e(t)$, and therefore, we report the value of $R_e(t)$ only at the start of each constant-transmission interval. The first lockdown significantly reduced the transmission rate. As

**Table 5. Posterior estimates for effective, $R_e(t)$, and control, $R_c(t)$ reproduction numbers during 2020.** Wave-one transmission rate estimates use data captured during the first wave only, whereas wave-two uses rates were estimated using data captured from the whole epidemic (see main text for further details). The final interval ended on 31st December 2020.

| Date of change | $R_e(t)^1$ Wave 1 | $R_c(t)^2$ Wave 1 | $R_e(t)^1$ Wave 2 | $R_c(t)^2$ Wave 2 |
|---|---|---|---|---|
| 31st January 2020 | 5.87 (5.32, 6.54)^ | 5.87 (5.32, 6.54)^ | 5.44 (4.51, 6.35)*^ | 5.44 (4.51, 6.35)*^ |
| 13th March 2020 | 3.02 (2.91, 3.12) | 3.02 (2.91, 3.12) | 2.84 (2.64, 3.02)* | 2.84 (2.64, 3.02)* |
| 24th March 2020 | 0.67 (0.65, 0.68) | 0.67 (0.65, 0.68) | 0.78 (0.76, 0.81)* | 0.78 (0.76, 0.81)* |
| 11th April 2020 | 0.81 (0.80, 0.81) | 0.81 (0.80, 0.81) | 0.80 (0.79, 0.80)* | 0.80 (0.79, 0.80)* |
| 10th August 2020 | 1.02 (0.88, 1.17) | 1.18 (1.02, 1.36) | NA | NA |
| 15th August 2020 | NA | NA | 1.62 (1.60, 1.65) | 1.71 (1.68, 1.74) |
| 6th September 2020 | NA | NA | 1.48 (1.47, 1.49) | 1.57 (1.55, 1.68) |
| 14th October 2020 | NA | NA | 1.23 (1.22, 1.24) | 1.30 (1.29, 1.31) |
| 5th November 2020 | NA | NA | 0.80 (0.78, 0.83) | 0.85 (0.83, 0.88) |
| 18th November 2020 | NA | NA | 1.15 (1.13, 1.17) | 1.23 (1.21, 1.25) |
| 10th December 2020 | NA | NA | 1.43 (1.39, 1.46) | 1.54 (1.50, 1.57) |

[1] $R_e(t)$ estimates are given for the new transmission after a date of change. Early $R_e(t)$ estimates do not substantially differ from $R_c(t)$ estimates due in negligible susceptible depletion.

[2] $R_c(t)$ estimates are given for the interval beginning at the date of change until the next date of change.

^ Based on very few data points, since the data starts on 1st March 2020, so may be unreliable. See trace plots (Fig A in S1 Supplementary Material) for poor identifiability of the initial growth rate.

*Based only on admissions rather than all four data streams so may be less reliable.

lockdown went on, $R_e(t)$ increased slightly, as indicated by the transmission rate change on 11th April 2020. Over August, transmission increased, bringing $R_e(t)$ above 1. This growth continued until further interventions were brought in with the local tier system. This reduced the transmission rate, likely driven by the effectiveness of the tier 3 interventions in the North West. Finally, the second lockdown brought transmission down across the whole of England, bringing again $R_e(t)$ below 1. Note that, using this model the initial reproduction number is not reliably constrained, since there are very few data points informing the initial transmission rate. This lack of identifiability is reflected in the MCMC trace plots (Fig A in S1 Supplementary Material).

## 2.4. Short-term forecasts were accurate unless transmission rates changed markedly during the forecasting window

To evaluate the performance of EpiBeds as a tool for real-time monitoring of the evolving epidemic in England, we performed two-week projections made on days 1 and 15 of each month, from March to December 2020, based on the data available at that time. We illustrate these projections in Fig 4, superimposed to the complete data for both waves. The posterior parameter estimates vary at every projection due to the additional data at each successive time point. We do not report the specific parameter estimates from each model fitting, but only the projections for the data streams. See Section SM.1.3.4 in S1 Supplementary Material for details on the setup when generating these results.

For the first forecasts (start date 1st April 2020), a transmission change was added on 24th March 2020 to allow EpiBeds to adjust transmission based on lockdown. Such a short fitting window resulted in large uncertainty, with both growing and declining epidemics falling

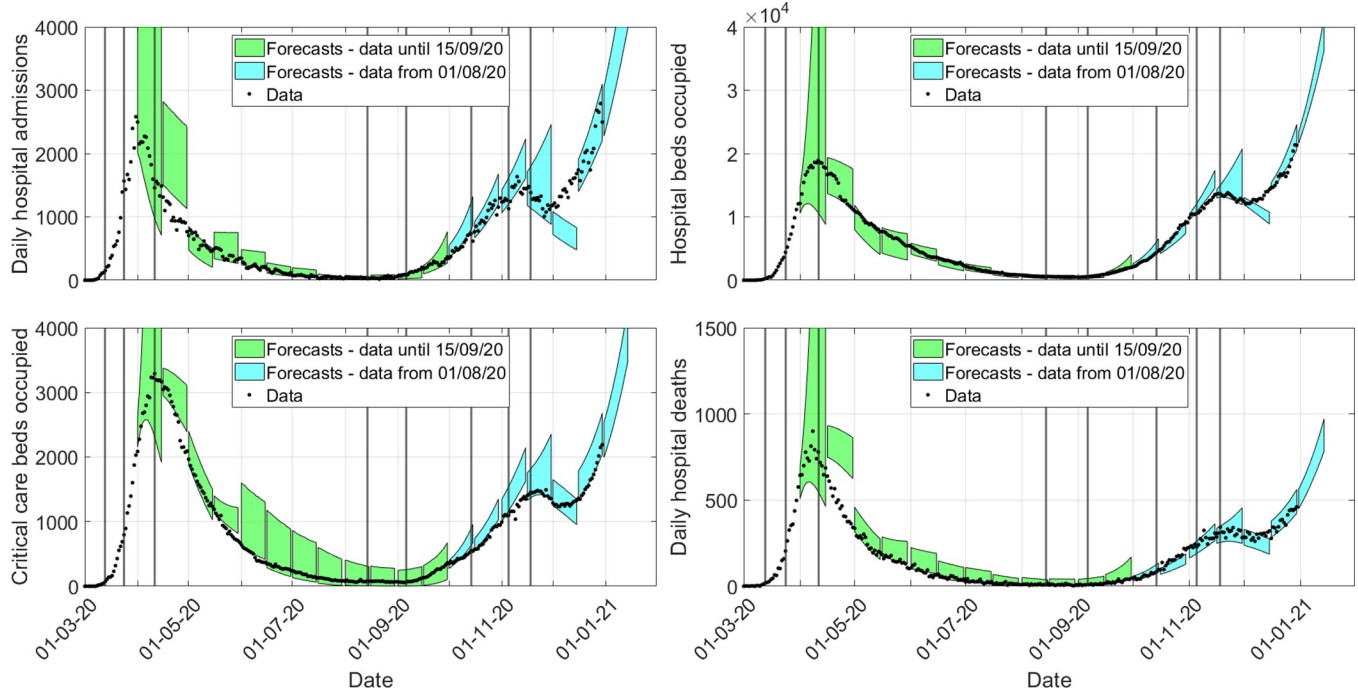

**Fig 4. England hospital forecasts.** Green shaded regions are the 90% prediction intervals from forecasts up to 15th September 2020. Blue shaded regions are the 90% prediction intervals from forecasts after 1st October 2020 (using data from 1st August 2020). Vertical black lines mark where major transmission changes occur, with changes in trajectory only manifesting after a delay that is data stream dependent. The y-axis is truncated to aid visibility, though a few forecast regions do exceed the y-limit.

within the 90% prediction interval. By the 15th April 2020 forecast, a peak had been observed in the admissions data, but EpiBeds was unable to reconcile the four data streams, which resulted in the forecasts underestimating the reduction in the transmission rate and overshooting the data. This poor performance could be driven by multiple factors, such as challenges with estimating length of stay early in the pandemic (Sections SM.1.2 and SM.3 in S1 Supplementary Material), changing demographics after entering the first lockdown, and data quality issues in some of the data streams (Section SM.1.1 in S1 Supplementary Material). After this point, forecasts remained reliable into the summer.

As transmission started to rise again, EpiBeds was able to accurately forecast the rise in all four data streams. However, throughout September and October, there was a demographic change, from younger to older age groups. This led to the ICU probability gradually declining and the mortality rate increasing, and the forecasts overestimated and underestimated, respectively, these two data streams. In November, the demographic distribution of cases stabilised, and EpiBeds was able to reconcile all four data streams. Noticeably, the 1st December 2020 forecast completely missed the trend in the data. This was partly to be expected since 2nd December 2020 marked the end of the second England-wide lockdown, and prior to this transmission rates were also likely to have been increasing due to behaviour changes and the emergence of the more transmissible Alpha variant [34].

Overall, 77% of data points, across all 4 data streams, fell within the 90% prediction intervals admissions 76%; hospital beds 80%; ICU beds 73%; deaths 80%). In many cases when data points fall out of the 90% prediction interval occur where an intervention was introduced during the forecasting window. Others potentially arise from data quality issues between the data streams, particularly during the first wave. Overall, this shows reasonably good model performance, and in practice throughout the pandemic EpiBeds has provided reliable forecasts in all regions where it was used. Our results highlight the context dependence of model performance, with lower predictive ability when transmission rates change frequently, and conversely greatly predictive ability when transmission rates are relatively stable.

## 3. Discussion

To make short-term predictions for the flow of patients through hospitals we developed EpiBeds, a compartmental model tailored to available line list data. The explicit inclusion of compartments depending on patient outcomes enabled the optimal use of available data whilst keeping model complexity low. By fitting the model to hospital occupancy data, we estimated the proportion of patients entering each hospital pathway, generated short-term hospital occupancy predictions, and helped inform management of hospital caseloads. Using the model, we were also able to estimate the effective and control reproduction numbers during different periods of the epidemic, corresponding to substantial changes in the hospital trends driven by major policy changes, the emergence of new variants, and seasonal effects. As well determining changes in the reproduction number during the 2020 epidemic in England, which largely corresponded to changes in policy, we also captured the greater proportion of hospitalised patients recovering between the first and second waves.

We validated the short-term forecasting performance of EpiBeds by generating 14-day forecasts using data available at the start and midpoint of each month. Due to the potentially fast-growing nature of COVID-19 outbreaks [18] about:blank, and the limited duration of most interventions, long-term forecasting is limited, since conditions are likely to have changed between the production of the forecast and reaching the forecast horizon. Because of the delay of a few weeks between the implementation of interventions and their effects on hospital admissions [18,35], short-term forecasts of a few weeks should not be significantly affected,

and are valuable planning tools for hospitals and health services. Most of our forecasts captured the data within the 90% prediction interval of the forecasts, demonstrating the reliability of EpiBeds for providing short-term hospital flow predictions. When transmission rates were stable, forecasting accuracy was particularly high. However, large changes in transmission rates, for example due to major policy changes and the emergence the Alpha variant, reduced the forecasting accuracy. Data quality issues can also affect predictions, and this likely contributed to some of the forecasting inaccuracies we observed during the first wave.

EpiBeds was developed specifically to provide predictions of hospital occupancy and designed to maximise the information in available data whilst minimising the inclusion of unsupported assumptions. For this reason, the model is not structured by sex or age, or other comorbidities such as heart failure or chronic kidney disease, even though these are known to affect disease severity [12,36]. The SITREP does not include sex as a category and does not include age for all data streams (particularly when the model was first developed). As epidemics progress, the communities in which the virus circulates may change, which in turn could affect how individuals progress through hospital pathways, such as the probability of entering ICU if critically ill. This emphasises the need for the consistent reporting of high-quality data so that estimates can be continuously updated, resulting in more accurate forecasts. To account for demographic changes, as well as potential improvements in treatments, we fitted the parameters for the second wave independently of the first wave.

The structure of EpiBeds makes use of the fact that the delay distributions were approximately Erlang distributed, so that they can easily be approximated by a series of ODEs. It would be possible to instead write the model in terms of delay equations, but the ODE approach leads to significantly reduced computational cost, which is essential for a modelling product that may need to be run multiple times per week.

A limitation of the current framework is the assumption of complete immunity. For the time period considered, this is unlikely to have affected the results. However, with mass vaccination, immune waning and immune escape variants, more complexity may be required to capture long term dynamics. To address this, vaccinated compartments, variants, and immune waning could be added to the model. However, over the short time scales of projections considered, population immunity is unlikely to have a major influence on the dynamics, which are mostly driven by recent trends in the data.

Our model differs from more conventional compartmental models by defining compartments based not only on current status, but on future outcome, making it more closely aligned to the data. This alignment to data, and its relative simplicity, means EpiBeds can be used to make short-term predictions in different settings, as well as used as a framework to develop short-term forecasts in the case of new outbreaks. Moreover, the minimal complexity of EpiBeds makes it easy to identify the cause of model fitting issues, including lack of identifiability of the patient outcome probabilities without strongly informative priors and temporal changes in the relationships between the different data streams, and makes both model behaviour and model limitations transparent. We deem these to be key reasons to advocate for the use of simple models. Here, we fitted EpiBeds to hospital data for England, but it can readily be applied to other geographies. For example, as part of the COVID-19 response, we used it to generate forecasts for Scotland, Wales, Northern Ireland, and the United Kingdom, as well as for smaller English regions.

## 4. Methods

### M.1 The ODE compartmental model for hospital flow

The structure of the ODE model was informed by the delay distributions (Section 2.1 and Section SM.1.2 in S1 Supplementary Material). In an ODE compartmental model with constant

progression rates, the permanence waiting times in each compartment are exponentially distributed with mean equal to the inverse of the rate. Since all the hospital length of stay distributions were approximately exponential (because the mean is similar to the standard deviation), we modelled the hospital compartments using constant rates. For the background epidemic, we represented the Gamma-distributed incubation period (shape parameter 3, [4]) and the Gamma-distributed time between symptom onset to hospitalisation (shape parameter 2, [5]), by using three and two compartments in sequence. This is a way of representing Gamma-distributed permanence times by using the Erlang distribution. Specifically, an Erlang distribution with shape parameter $n$ corresponds to the sum of $n$ independent and identically distributed exponential distributions [6]. If all rates are equal to $r$, the mean permanence time in a sequence of $n$ compartments will be Erlang distributed with mean $n/r$ and shape parameter $n$ [7]. This results in EpiBeds described by the flowchart in Fig 1 and by the following system of ODE's (where the time index in all parameters and variables has been dropped for brevity)

$$\frac{dS}{dt} = -\lambda$$

$$\frac{dE_{S_1}}{dt} = (1 - p_A)\lambda - 3r_E E_{S_1}$$

$$\frac{dE_{S_2}}{dt} = 3r_E E_{S_1} - 3r_E E_{S_2}$$

$$\frac{dE_{A_1}}{dt} = p_A \lambda - 3r_E E_{A_1}$$

$$\frac{dE_{A_2}}{dt} = 3r_E E_{A_1} - 3r_E E_{A_2}$$

$$\frac{dI_s}{dt} = 3r_E E_{S_2} = 3r_E I_S$$

$$\frac{dI_A}{dt} = 3r_E E_{A_2} - 3r_E I_A$$

$$\frac{dL_R}{dt} = 3r_E I_S (1 - p_H) - r_{LH} L_R$$

$$\frac{dA_R}{dt} = 3r_E I_A - r_{AR} A_R$$

$$\frac{dL_{H_1}}{dt} = 3r_E I_S p_H - 2r_{LH} L_{H_1}$$

$$\frac{dL_{H_2}}{dt} = 2r_{LH} L_{H_1} - 2r_{LH} L_{H_2}$$

$$\frac{dH_D}{dt} = 2r_{LH} L_{H_2} p_T - r_{HD} H_D$$

$$\frac{dM_R}{dt} = r_{CM}C_M - r_{MR}M_R$$

$$\frac{dD}{dt} = r_{HD}H_D + r_{CD}C_D$$

$$\frac{dN}{dt} = -r_{HD}H_D - r_{CD}C_D$$

$$\frac{dR}{dt} = r_{HR}H_R + r_{MR}M_R + r_{LR}L_R + r_{AR}A_R$$

where

$$\lambda = S\beta(f(I_A + A_R) + I_S + L_R + L_{H_1} + L_{H_2}).$$

Here, $f$ is the reduction in transmission for asymptomatic cases, which is taken to be $f = 0.25$.

From the solution to the ordinary differential equations, the control and effective reproduction numbers can be calculated. The control reproduction number is given by

$$R_c(t) = \beta(t)k,$$

where

$$k = (1 - p_A)\left(\frac{1}{3r_E} + \frac{p_H}{r_{LH}} + \frac{1 - p_H}{r_{LR}}\right) + p_A f\left(\frac{1}{3r_E} + \frac{1}{r_{AR}}\right).$$

From the control reproduction number, the effective reproduction number can be calculated as

$$R_e(t) = R_c(t)\frac{S(t)}{N(t)}.$$

## M.2 Markov Chain Monte Carlo (MCMC)

To fit the ODE model to data, we generated a likelihood function which we then optimised using MCMC. Specifically, adding Negative Binomial noise to the ODEs describing the model enabled us to calculate a likelihood function for observing the data given our model parameters. This is based on the probability that the deviation between our model and the data can be explained by noise. For each of the four data streams we constructed a likelihood function, which were then multiplied together to build the overall likelihood function. In addition, we included an informative prior for the probability of dying in ICU, $p_D$, giving an overall likelihood function:

$$L = \sum \ln\left(f\left(d_A, \frac{y_A}{\sigma_A - 1}, \frac{1}{\sigma_A}\right)\right)$$

$$+ \sum \ln\left(f\left(d_B, \frac{y_B}{\sigma_B - 1}, \frac{1}{\sigma_B}\right)\right)$$

$$+ \sum \ln\left( f\left( d_C, \frac{y_C}{\sigma_C - 1}, \frac{1}{\sigma_C}\right)\right)$$

$$+ \sum \ln\left( f\left( d_D, \frac{y_D}{\sigma_D - 1}, \frac{1}{\sigma_D}\right)\right)$$

$$- \left( \frac{1}{2}\ln(2\pi\sigma_{prior}^2)\right) - \frac{1}{(2\sigma_{prior}^2)(p_D - \mu_{prior})^2}$$

Where $A$, $B$, $C$ and $D$ refer to the four different data streams fitted and $\sigma$ is the overdispersion parameter of the Negative Binomial observation noise, $d$ is the data, $y$ is the solution to the ODEs, $\mu_{prior}$ is the mean prior estimate of $p_D$ and $\sigma_{prior}$ is the standard deviation of the prior $p_D$ estimate. The continuous variables $y$ are defined as

$$y_A = r_{LH}L_{H_2}$$

$$y_B = H_D + H_C + H_R + C_D + C_M + M_R$$

$$y_C = C_D + C_M$$

$$y_D = r_{HD}H_D + r_{CD}C_M$$

and were evaluated at each day for which a data point $d$ was available. The sums are over all days for which data is available. Adding an informative prior for $p_D$ was required to constrain the values for $p_C$ and $p_T$.

To fit the model, we manually tuned a random walk MCMC algorithm implemented in Julia, with the input data depending on whether the first wave or second wave was being fitted. We start the epidemic on 20[th] January 2020, with $I_0$ initial cases in the $E_A$ and $E_S$ states. This allowed sufficient time for the other compartments to reach roughly stable proportions before the first data point on 1[st] March 2020. Prior values for EpiBeds parameters are specified as described in Sections SM.1.3.1 –SM.1.3.2 in S1 Supplementary Material, coupled with initial conditions for the free parameters with uninformative priors. The ODE was then solved for the input parameters, generating the time-series output that are added to the likelihood functions. Based on these likelihoods, the parameter values are scored and resampled, allowing EpiBeds to explore the parameter space. Code for simulating EpiBeds, and generating the scenarios shown in the paper, are available at [11], along with trace plots for all MCMC results included in this paper. Unfortunately, input data cannot be shared, since this was provided through a data sharing agreement, but similar publicly available data are available at [2].

When fitting the data, we considered the first and second waves separately. Due to changes in length of stay and patient outcomes over time, we cannot fit a single set of parameters over the whole pandemic. To fit the first wave of the epidemic, we used all four data streams, using data starting on 1[st] March 2020. When fitting the second wave, we removed beds, ICU, and deaths data prior to 1[st] August 2020. Prior to this date, EpiBeds is only constrained by the hospital admissions data, and only the first term of the likelihood (which does not depend on the outcome probabilities $p_C$, $p_T$ and $p_D$) is used. After 1[st] August 2020, we introduce the other three data streams and compute the other likelihood terms. This then constrains the probabilities to fit the relationship between these data streams in the second wave.

## Supporting information

**S1 Supplementary Material. Additional details describing the methods for EpiBeds.** Extra figures supporting the narrative. Additional results detailing the input parameters used for the performance evaluation.
(DOCX)

## Acknowledgments

The authors would like to thank colleagues in SPI-M-O and JUNIPER consortium for various discussions around hospital modelling and forecasting.

## Author Contributions

**Conceptualization:** Christopher E. Overton, Lorenzo Pellis, Katrina A. Lythgoe.

**Data curation:** Christopher E. Overton, Lorenzo Pellis, Helena B. Stage, Francesca Scarabel, Anel Nurtay, Katrina A. Lythgoe.

**Formal analysis:** Christopher E. Overton, Lorenzo Pellis, Helena B. Stage, Francesca Scarabel, Joshua Burton, Filippo Pagani, Katrina A. Lythgoe.

**Methodology:** Christopher E. Overton, Lorenzo Pellis, Francesca Scarabel, Joshua Burton, Ian Hall, Thomas A. House, Chris Jewell, Filippo Pagani, Katrina A. Lythgoe.

**Software:** Christopher E. Overton, Lorenzo Pellis, Francesca Scarabel, Joshua Burton, Filippo Pagani.

**Visualization:** Christopher E. Overton, Lorenzo Pellis, Helena B. Stage, Katrina A. Lythgoe.

**Writing – original draft:** Christopher E. Overton, Lorenzo Pellis, Helena B. Stage, Francesca Scarabel, Christophe Fraser, Anel Nurtay, Katrina A. Lythgoe.

**Writing – review & editing:** Christopher E. Overton, Lorenzo Pellis, Helena B. Stage, Francesca Scarabel, Anel Nurtay, Katrina A. Lythgoe.

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
