## [Decision Letter · Decision Letter 0]

4 Feb 2022

Dear Dr Overton,

Thank you very much for submitting your manuscript "EpiBeds: Data informed modelling of the COVID-19 hospital burden in England" for consideration at PLOS Computational Biology.

As with all papers reviewed by the journal, your manuscript was reviewed by members of the editorial board and by several independent reviewers. In light of the reviews (below this email), we would like to invite the resubmission of a significantly-revised version that takes into account the reviewers' comments.

We cannot make any decision about publication until we have seen the revised manuscript and your response to the reviewers' comments. Your revised manuscript is also likely to be sent to reviewers for further evaluation.

Sincerely,

Claudio José Struchiner, M.D., Sc.D.

Associate Editor

PLOS Computational Biology

Virginia Pitzer

Deputy Editor-in-Chief

PLOS Computational Biology

Reviewer's Responses to Questions

**Comments to the Authors:**

Reviewer #1: The summary of my review and comments addressed to authors are uploaded as an attachment.

Reviewer #2: Review of the manuscript 'EpiBeds: Data informed modelling of the COVID-19 hospital burden in England' by Christopher Overton and colleagues, submitted to PLOS Computational Biology.

Summary

In this manuscript, Overton and colleagues fit a transmission model for SARS-CoV-2 in England together with a hospital progression model to national-level data on hospital admissions, hospital occupancy (including ICUs), length of stay distributions in the hospital and in the intensive care, and hospital discharge and death data. Main goal is to obtain estimates of the durations in the various compartments in hospitals in England. The authors argue that the main value of the model is that has provided weekly forecasts of bed occupancy and admissions during the early stage of SARS-CoV-2 pandemic, and in addition suggest that the model is easily be adapted to apply to different pathogens and countries.

Evaluation

Overall, the methods are laid out clearly, the model code is publicly available, and I have no doubt that the analyses are sound. Also, I believe that the analyses may have helped the English government and public health bodies to anticipate hospital demand. I do have some reservations as to what the scientific novelty is of the analyses presented in the manuscript is. It is true that the within-hospital progression model is somewhat more complex than most (perhaps all) other transmission models that have been fitted to hospital data, but many aspects that could have made this manuscript stand out are missing. For instance, (1) all analyses are performed for national level data and as far as I can see no attempt has been made to include analyses at the regional of hospital level. This is unfortunate as the national-level data are the resultant of the superposition of local epidemics. (2) As far as I can see no attempt has been made to analyze and fit age-stratified models to age-stratified data, even though it is known that hospitalization rates are strongly age-dependent while age-stratified incidence has also changed quite a bit during the pandemic. This problem is now partially solved by defining different periods for the analyses, but it would have been nice if everything would have been fitted in one go with an age-stratified transmission model. (3) Hardly any formal (in the statistical sense) effort been undertaken to evaluate predictive performance of the model (e.g., using leave-one-out cross-validation), and I did not spot any formal attempt of model selection. In essence, the authors use a single model and rely on visualizations to spot where data and model are congruent. In all, I do not suggest that the authors should actually address the above points by adding more models and analyses, but it does lessen my enthusiasm for the manuscript. I suggest the authors put more effort in a critical evaluation of their model in the discussion, and perhaps could add something on cross-validation and model selection in the main analyses.

Specific comments/suggestions

-intro, second paragraph. please add that assessment of the current situation (i.e nowcasting) is a problem in itself. Also add in the discussion how your results are affected if cases (by admission date) are only complete after some time (two weeks? is this a problem?)

-Figure 1. I found this figure not very appealing visually, and the legend difficult to follow.

-"With the foresight of formulating ...". It is of course quite convenient that delay distributions are apparently well-described by exponential distributions. It is, however, not difficult to rewrite the model from the hospital compartment (L_H) onwards in terms of delay equations. Please discuss and elaborate in the Discussion.

-Figure 2. I found this somewhat superfluous, especially the distinction between i. and ii.

-"The structure for the generalised ...". Please mention once that formally these are a specific class of exponential distributions, i.e. Erlang distributions.

-"This level of accuracy is sufficient since ...". This strikes me as an unwarranted claim. Please remoce or explain in detail.

-Section 2.3.1. Please provide rationale WHY most parameters are fixed while only a small number is estimated. Especially, as in the Bayesian context you have the flexibility to add (weakly) informative priors. Why did you include a highly informative prior for p_D but not for the other p parameters?

-Figure 4, bottom left. There is a systematic deviation from the posterior median. Explain in intuitive terms? Also, are bands actual Bayesian prediction intervals, or CrIs?

**Have the authors made all data and (if applicable) computational code underlying the findings in their manuscript fully available?**

Reviewer #1: None

Reviewer #2: Yes

PLOS authors have the option to publish the peer review history of their article (what does this mean?). If published, this will include your full peer review and any attached files.

Reviewer #1: **Yes: **Quentin J Leclerc

Reviewer #2: No
---

## [Decision Letter · Decision Letter 1]

18 Jul 2022

Dear Dr Overton,

We are pleased to inform you that your manuscript 'EpiBeds: Data informed modelling of the COVID-19 hospital burden in England' has been provisionally accepted for publication in PLOS Computational Biology.

Best regards,

Claudio José Struchiner, M.D., Sc.D.

Associate Editor

PLOS Computational Biology

Virginia Pitzer

Deputy Editor-in-Chief

PLOS Computational Biology

Reviewer's Responses to Questions

**Comments to the Authors:**

Reviewer #1: I would like to congratulate the authors again for this interesting work, and am glad they found my comments useful. I am happy with the revisions made to the manuscript. The paper is now much more streamlined and easier to read. I do not have any further comments or suggestions for the authors.

**Have the authors made all data and (if applicable) computational code underlying the findings in their manuscript fully available?**

Reviewer #1: None

PLOS authors have the option to publish the peer review history of their article (what does this mean?). If published, this will include your full peer review and any attached files.

Reviewer #1: **Yes: **Quentin J Leclerc

---

## [Editor Report · Acceptance letter]

31 Aug 2022

PCOMPBIOL-D-21-02185R1 

EpiBeds: Data informed modelling of the COVID-19 hospital burden in England

Dear Dr Overton,

I am pleased to inform you that your manuscript has been formally accepted for publication in PLOS Computational Biology. Your manuscript is now with our production department and you will be notified of the publication date in due course.

With kind regards,

Anita Estes
